# Tumor and Tumor-like Lesions in Red Foxes (*Vulpes vulpes*) from Croatia

**DOI:** 10.3390/ani14040558

**Published:** 2024-02-07

**Authors:** Šimun Naletilić, Ivan-Conrado Šoštarić-Zuckermann, Željko Mihaljević

**Affiliations:** 1Croatian Veterinary Institute, Savska Cesta 143, 10000 Zagreb, Croatia; naletilic@veinst.hr (Š.N.); miha@veinst.hr (Ž.M.); 2Faculty of Veterinary Medicine, University of Zagreb, Heinzelova 55, 10000 Zagreb, Croatia

**Keywords:** red fox, *Vulpes vulpes*, tumor, tumor-like lesion, Croatia

## Abstract

**Simple Summary:**

The red fox (*Vulpes vulpes*), a member of the Canidae family, is found on all continents except Antarctica. In Croatia, red foxes are indigenous wild species that inhabit both the coastal and continental regions of the country. Neoplastic diseases are the leading cause of mortality in pet dogs, but wild members of the Canidae family can also develop tumors. Tumors in non-domestic canids are scarce; however, the true prevalence and diversity of these tumors are likely underestimated due to limited research. So far, only a limited number of tumors have been observed among the red fox population, either in their natural habitat or in captivity. The lack of wide-scale research on the incidence of neoplasia in the population of wild carnivora species in Croatia underscores the need for further investigation to enhance our understanding of tumors in foxes. In this study, conducted on a sample of 1890 red foxes, five tumor or tumor-like lesions were identified. These included two Meibomian gland adenomas, two collagenous hamartomas, and one teratoma. This research is the first of this kind in Croatia, shedding light on the occurrence and types of tumors in the red fox population.

**Abstract:**

The red fox, found on all continents except Antarctica, occupies diverse habitats. In Croatia, it is an indigenous wild species with a population density of 0.7 animals per square kilometer. While tumors in wild animals from the Canidae family are scarce, the true prevalence and diversity of tumors are likely underestimated due to limited research. So far, a limited number of tumors have been observed among the red fox population, either in their natural habitat or in captivity. As part of the National Rabies Control Program, we examined 1890 red fox carcasses over a four-year period. Our focus was on identifying abnormalities on the skin and internal organs that suggest potential neoplastic proliferation. Five red foxes, three males and two females, were found to have growths resembling potential tumors. Their age distribution spanned from 2 to 7 years. Microscopic investigation revealed two collagenous hamartomas, two Meibomian gland adenomas, and one intra-abdominal teratoma within a cryptorchid testis. This retrospective study aims to provide a comprehensive description of tumor and tumor-like lesions observed in free-range red foxes from Croatia, marking the first research of its kind in Croatia.

## 1. Introduction

The Canidae family comprises approximately 35 species, including the red fox (*Vulpes vulpes*), which is considered to be the most geographically widespread member of this family [1,2]. They are found on all continents except Antarctica and occupy diverse habitats, including urban areas and natural habitats, owing to their remarkable ability to adapt to various environments and food resources [2,3]. Red foxes in Croatia are considered indigenous wild species, with a population density estimated at 0.7 animals per square kilometer [4]. They are found in a wide range of habitats, including coastal and continental areas, and are known to inhabit urban areas as well [3,4]. Due to their proximity to humans and domestic animals, such as dogs, they can potentially act as reservoirs for infectious and zoonotic agents [5]. Orphanhood is the most common cause of death in young foxes, while other categories of foxes most frequently die as a result of trauma or mange [6,7]. While these factors contribute to the mortality of different age categories of foxes, it is noteworthy that neoplastic diseases emerge as the primary cause of death in dogs in developed countries [8]. Considering these factors, especially given the shared family classification of foxes and dogs, it is useful to investigate the prevalence of tumors in the red fox population. The occurrence of neoplastic disease in wildlife has been observed to lead to substantial rates of morbidity and mortality [9]. Within nature, a small number of neoplasms occur that specifically target certain species of wild animals, thereby having a significant influence on the species’ overall survival. One example is the devil facial tumor disease, a highly aggressive, transmissible, and inevitably lethal cancer that affects Tasmanian devils (*Sarcophilus harrisii*), resulting in an alarming 80 percent decline in their population [10,11]. Devil facial tumor disease is a naturally transmitted tumor that is spread through bites. It primarily affects the facial region and commonly metastasizes to nearby lymph nodes, lungs, and kidneys [11]. The second tumor that affects a significant portion of the fox population is adenocarcinoma of the ceruminous glands in the ear of endangered foxes (*Urocyon littoralis catalinae*) on Santa Catalina Island. This tumor affects approximately 48.9 to 52.2 percent of the fox population on the island. Its occurrence is linked to a chronic and long-term infection with ear mites, which leads to an extremely severe inflammatory reaction and hyperplasia in response to these mites [12]. The prevalence of cancer in captive animals has been noted to be greater in comparison to their counterparts in the wild [13]. The extended lifespan of captive animals can be attributed to improved veterinary management and care [13]. Tumors occurring in non-domestic canids are scarce, and the true prevalence and diversity of tumors are likely underestimated due to limited research [1]. It is generally accepted that the spectrum of tumor types seen in domestic dogs may also be present in wild species [1]. So far, only a limited number of neoplasms have been observed among the red fox population, both in their natural habitat and in captive settings, and there is a dearth of information regarding the prevalence of these tumors within this particular wild species. The majority of the documented cases consist of isolated reports of incidentally discovered tumors, with only a rare instance of their inclusion in comprehensive systematic investigations. Within 873 examined red foxes from Denmark, authors found only one tumor, which was apparently an insulin-producing islet cell tumor [14]. No instances of neoplasms were detected in a study conducted on a population of 530 red foxes in Germany [15]. A study conducted in Austria in 1999 identified the presence of mammary adenocarcinoma in a female red fox [15]. A publication in Italy documented a case of renal cell carcinoma in a red fox [16]. The occurrence of thyroid C-cell carcinoma with amyloid was documented in a 15-year-old red fox bred at Akita Ohmoriyama Zoo [17]. Individual case reports described the occurrence of adenosquamous carcinoma [18], seminoma, teratoma [19], and a lipoma [20]. There are also reported cases of tumors in the fennec fox (*Vulpes zerda*) and swift fox (*Vulpes velox*) that belong to the genus *Vulpes*. The fennec fox population has a significant incidence of hepatocellular carcinoma [21]. Furthermore, individual cases of tumors in the fennec fox have been documented, such as oral melanoma, thyroid gland adenoma [22], and nephroblastoma [23]. In addition, a single case of intestinal adenocarcinoma in a swift fox was also documented [24]. There is a lack of comparable research on the population of wild carnivora species in Croatia, but there have been two retrospective studies on the incidence of canine tumors in the dog population of Croatia. In the first study, an estimation of tumor prevalence was conducted in the overall canine population. The findings revealed that most of the tumors were malignant, with the primary sites of origin being the skin and subcutaneous tissues. This was followed by tumors originating from the mammary glands and the genital region. The prevailing tumor types identified in the study were mammary tubulopapillary carcinoma, mast cell tumor, and fibrosarcoma [25]. The second study exclusively examined tumors found in indigenous Croatian dog breeds. The findings indicate that a majority of these tumors were malignant, with a higher prevalence observed in the skin, mammary glands, and hemolymphatic system [26].

Taking into consideration the aforementioned observations, the objective of this retrospective study is to provide a comprehensive description of tumor and tumor-like lesions observed in free-range red foxes from the central region of Croatia. This research, being the first of its kind conducted in Croatia and in this part of Europe, aims to enhance our understanding of tumor prevalence in wildlife and contribute valuable insights for conservation strategies.

## 2. Materials and Methods

A systematic examination was conducted on red fox (*Vulpes vulpes*) carcasses submitted to the Laboratory for Pathology at the Croatian Veterinary Institute in Zagreb over a four-year period, spanning from June 2019 to June 2023. These carcasses were obtained as part of the rabies control program mandated by the Croatian Ministry of Agriculture, Veterinary, and Food Safety Directorate. A methodical and thorough protocol was followed during the necropsy of all submitted carcasses. This protocol included a thorough examination of the skin and its formations, the external genitalia, the mucous membranes, and a systematic examination of the internal organs, beginning with the oral cavity and continuing through the neck, chest, abdominal cavity, and head. During the examination of deceased red fox specimens, special emphasis was placed on the detection of potential tumors. Based on the visual characteristics and growth patterns, these abnormalities were classified as tumors and tumor-like or nontumorous. All these tumor or tumor-like lesions were meticulously documented through photography and subsequently completely extracted and placed in adequate containers for fixation solution in 10% neutral formalin for a period of 24 to 48 h. A standard protocol for routine histology was followed, including dehydration, embedding in paraffin blocks, and cutting serial 4 µm thick slices. The sections were stained with hematoxylin and eosin and prepared for examination under a microscope.

Additionally, the sex of each fox with a tumor or tumor-like lesion was noted, and the age of each delivered fox was estimated using the teeth, following a previously established protocol [27].

Using an Axio Imager.A2 microscope (Zeiss, Oberkochen, Germany), all histology slides were examined, and images were captured with a Digicyte BigEye microscope camera.

## 3. Results

A total of 1890 fox carcasses were examined between June 2019 and June 2023 as part of the Croatian Ministry of Agriculture, Veterinary and Food Safety Directorate’s rabies control program. There were eleven counties from which red foxes originated: Krapina-Zagorje, Varaždin, Međimurje, Karlovac, Lika-Senj, Sisak-Moslavina, Bjelovar-Bilogora, Brod-Posavina, Požega-Slavonia, and Dubrovnik-Neretva. Of all the foxes examined, 64% were killed during the legal hunt. In 32% of the foxes, trauma from a motor vehicle collision was the second cause of death. During the necropsy of only 4% of dead foxes discovered in public areas, it was found that the pathological alterations corresponded to infectious or parasitic diseases (Figure 1).

Most of the delivered foxes were between 1 and 3 years of age, but the range was from at least 1 month to more than 7 years (Figure 2).

Following a thorough examination as outlined in the materials and methods, tumor or tumor-like lesions were found in five foxes (0.2645%; CI 0.08595–0.61628). Only five red fox carcasses out of the total number of foxes examined had growths that were classified as tumors or tumor-like (Table 1). All foxes with detected tumors or tumor-like lesions were killed during the legal hunt.

Out of these five foxes, three were male, while the remaining two were female. The age distribution spanned from 2 to 7 years, with a mean age of 4.4 years, indicating a relatively young population. Within the five tumor/tumor-like lesions observed, four were located on the integumentary system, specifically in the regions encompassing the lower and upper eyelids, nose, and ventral abdomen. The fifth tumor was found within the abdominal cavity. Histological examination of the tumor/tumor-like lesions on the fox’s skin revealed two of these as Meibomian gland adenomas (Figure 3 and Figure 4), another two as collagenous hamartomas (Figure 5 and Figure 6), while the intra-abdominal outgrowth was diagnosed as a teratoma (Figure 7) within a cryptorchid testis.

The dermal structures present on the upper and lower eyelids exhibited a resemblance to a cauliflower, characterized by a polypoid morphology. On the cross section, these tumors were of gray to white color. The histological examination revealed a notable increase in the number of sebaceous glands, accompanied by a thin layer of non-proliferative cells, within a mild increase in the collagenous stroma. The presence of multifocal infiltration of inflammatory cells was also observed.

Collagenous mesenchymal structures were observed on the abdominal and nasal regions of two red fox specimens. The abdominal tumor-like lesion was characterized by a circular shape, accompanied by a stalk that connected the outgrowth with the abdominal skin. The observed cross-section exhibited a white-gray color, characterized by a connective tissue framework. The nasal outgrowth exhibited a prominent growth at the apex of the nose, resembling a horn. Upon closer inspection, the cross-sectional view revealed a white-gray color and a connective tissue arrangement. From a histological perspective, it can be observed that both tumor-like lesions consisted of excessive amounts of dermal collagen. The arrangement of the fiber pattern had a resemblance to the organization of collagen bundles in the dermis.

A teratoma was detected in a juvenile male with undescended testicles who was shot during legal hunting. The carcass exhibited moderate cachexia. There was a single testicle located within the scrotum (a unilateral cryptorchid). Within the abdominal cavity, there was a mass that had an irregularly circular shape and a diameter of approximately 22 cm. This formation was connected to the mesentery through adhesions. The mass encompassed 25% of the abdominal cavity. The cross-section exhibited a central compact region, which was yellow and consisted of a connective tissue structure with multiple localized areas of mineralization. There was a vascular region on both sides that contained multiple blood-filled spaces. From a histological point of view, we observed regions of cartilage formation with osteoid and multinuclear osteoclasts. There were areas of connective tissue within the tumor’s vascular region that frequently contained elongated, prism-shaped epithelial cells with well-developed cilia on their surfaces. The identification of two distinct cell types, each corresponding to one of the embryonic germ layers, led to the diagnosis of teratoma.

## 4. Discussion

Foxes, belong to the canid family and are the most numerous predator species found throughout Croatia [4]. According to many authors, they serve as potential reservoirs for a wide range of infectious diseases and zoonotic pathogens, including the rabies virus, *Trichinella* spp., *Echinococcus* spp., *Leptospira* [4,5,28,29], and *Salmonella* spp. [30]. The primary cause of mortality among foxes is commonly attributed to instances of being orphaned (33%), followed by trauma (27%), and sarcoptic mange (17%) [6]. The examination of observed pathological alterations in the organ systems of foxes in Germany revealed that the gastrointestinal system is the most commonly affected, followed by the respiratory, genitourinary, cardiovascular, integumentary, nervous, and lastly, the musculoskeletal system [2]. Based on the findings of these studies, it has been observed that foxes, along with other animal species, possess the potential to acquire various diseases with diverse origins. Due to the nature of our research, which focused on identifying tumors in foxes obtained through hunting or found dead on roads, only a limited number of foxes died from natural causes. Therefore, we are unable to make comparisons or assess the prevailing natural causes of death in the fox population in Croatia.

Tumors pose a significant risk to wild animals, with certain types leading to elevated mortality rates among specific vertebrate populations [9,31]. Viruses are responsible for a significant portion of these tumors [31].

To date, no research has been conducted in Croatia on this matter, thus precluding any direct comparison of findings with existing studies on tumor occurrence in wild animals that belong to the family Canidae. However, we will proceed to compare the obtained results with established data on tumor prevalence in domestic dogs within Croatia, as well as with data on reported tumors in foxes.

This study represents the initial investigation carried out in Croatia, aiming to analyze tumor and tumor-like lesions in red foxes within this specific region of Europe. In our study, it was observed that among a sample size of 1890 foxes, just five individuals had the presence of a tumor or tumor-like lesion, which was a comparatively elevated prevalence when compared to other regions in Europe. The salient feature of these findings is that all identified lesions were observed in red foxes living in their natural habitat, and furthermore, all tumors were classified as benign. This stands in contrast to previous studies where the majority of tumors in foxes were found to be malignant [15,16,17,18,21,22,23,24]. This number of only five animals with a tumor or tumor-like lesion is relatively low when compared with the considerably high incidence of tumors in canines, where it stands as the primary cause of mortality [8,25,26]. Three individuals were identified as males, representing 60% of the animals, while two individuals were identified as females (40%). Although this is a very small sample size, this is consistent with previous studies in foxes, which have shown a higher prevalence of tumors in male animals compared to females. When considering only red foxes in the reported cases, there is a higher incidence of tumors in female red foxes. Also, these findings correspond with previous studies conducted on dogs [25,26]. Although the population under examination encompassed a wide age spectrum, ranging from a few months to over 7 years, it is noteworthy that the group of foxes with the tumor had a relatively young average age. In our study, a relatively young population (mean age 4.4 years) of foxes had tumors and tumor-like lesions, which contrasts findings in dogs, where tumors were predominantly observed in animals of middle or older age [8,25,26], but also most reports on tumor occurrences among foxes have highlighted a higher prevalence among older individuals, particularly those held in captivity [17,18,20,21,22,23,24]. The majority of these established tumorous lesions (80%) were found on the skin, while the remaining one was located in the abdomen (20%). This distribution aligns with findings from studies conducted on dogs, which have consistently reported a higher prevalence of skin tumors compared to tumors in other organ systems [25,26]. However, this does not align with the documented tumors in foxes, where only a single case of a cutaneous or subcutaneous tumor was described, this being subcutaneous lipoma [20].

In our study, it was observed that all skin tumors that were identified were benign tumors or tumor-like lesions, specifically meibomian gland adenoma and collagenous hamartoma. This corresponds with the previously documented data in foxes, where so far, no malignant skin tumors have been detected, and the only tumor detected was the previously mentioned lipoma case [20]. Interestingly, this contrasts with the data in dogs, where skin tumors are predominantly malignant [25,26].

Eyelid neoplasms are the predominant periocular tumors in elderly canines, with the majority being unilateral, benign, and slow-growing [32]. Meibomian gland adenoma arises from the tarsal glands (meibomian glands) on the inner aspect of the eyelid and is a frequently encountered benign neoplasm affecting the eyelids of canines [33]. Typically, they manifest as small, raised tumors with an irregular surface and a pink, grey, or black color [32,33], as in our cases. According to our research, solitary masses on the lower and upper eyelids represent 40% of all detected tumor lesions, or 50% of detected skin tumors.

Collagen hamartomas are common solitary and nodular tumor-like lesions in dogs [34,35]. Lesions can manifest on various regions of the body, although they are most commonly found on the head and limbs [34]. In the specific case of a red fox, we observed the presence of lesions on the nose, while in other instances, they were located in the caudal abdomen region. The histological findings in red foxes align with the hamartomas in dogs.

A tumor that was discovered inside the abdomen originated from an undescended testis in a 2-year-old fox. Cryptorchidism represents the prevailing anomaly in sexual development among juvenile males [36]. It occurs most often in stallions [37] but is also a frequent clinical finding in dogs [36]. Ninety percent of tumors of the male dog genital system are testicular tumors, which are the second most common type of tumor in male dogs [38]. The most prevalent type of tumor in the testis of dogs is Leydig cell tumor, but Sertoli cell tumor and seminoma are more prevalent in cryptorchids [36,39], which is not the case in this red fox where we detected testicular teratoma. Teratomas are uncommon tumors in dogs [40] that are formed of abnormal tissue originating from at least two, and frequently all three, germinal layers [37,41]. Typically, they are composed of histologically benign, well-differentiated tissue [42]. Macroscopically, this type of tumor is characterized by multiple cystic formations, often with the presence of hair and secretions containing mucoid or sebaceous substances, and solid masses of yellow and white composition consisting of fibrous, adipose, cartilaginous, and bony tissue [36], which corresponds to the macroscopic appearance of teratoma in our case. On histology, we detected tissues derived from two embryonic germ layers, which fulfills the criteria for the diagnosis of teratoma.

Arguably, the most influential factor of this study, skewing our results, was the young age of the examined fox population, which can be seen as a crucial limiting factor. The majority of foxes observed were either young individuals or mature juveniles, which contrasts with the data available for dogs, where the highest number of dogs affected by tumors are older animals. An additional aspect influencing the data is the inability to comprehensively study the entire fox population in Croatia, contrasting dogs, which are predominantly registered and supervised by their owners. The final factor that partially restricts the scope of the subject study is the potential occurrence of very small tumors either on the skin or within the internal organs, which might have been overlooked.

Despite the study’s limitations, it must be emphasized that a substantial sample of foxes was analyzed, offering valuable insights into tumor occurrence in this wild animal species that is difficult to study. Undoubtedly, this study represents the initial research of its kind in Croatia, necessitating a more comprehensive and methodical examination to acquire additional data that will address the deficiencies identified by this study.

## 5. Conclusions

This study marks the inaugural investigation of its kind in Croatia, presenting a unique modality to explore tumorous and tumor-like lesions in foxes within this specific region of Europe. It lays the groundwork for future surveillance of these lesions in foxes living in their natural habitat. The identification of a limited number of tumors is likely linked to the young age of all the animals involved, since it is widely acknowledged that tumorous lesions predominantly appear in older canines.

To improve the accuracy of the analysis, it is crucial to systematically monitor the elderly fox population over an extended period of time. This will contribute to a more comprehensive understanding of the prevalence and nature of these lesions in the long term.

## Figures and Tables

**Figure 1 animals-14-00558-f001:**
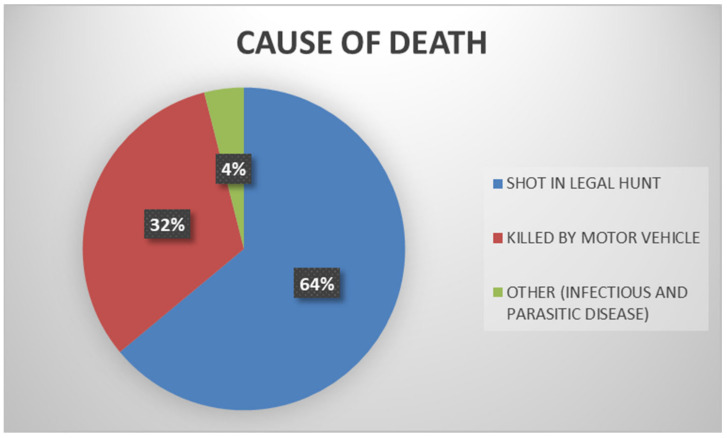
Causes of death in submitted foxes.

**Figure 2 animals-14-00558-f002:**
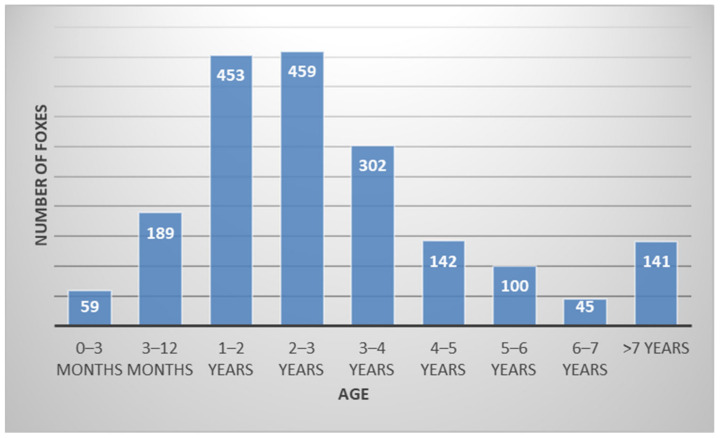
Age range of submitted foxes.

**Figure 3 animals-14-00558-f003:**
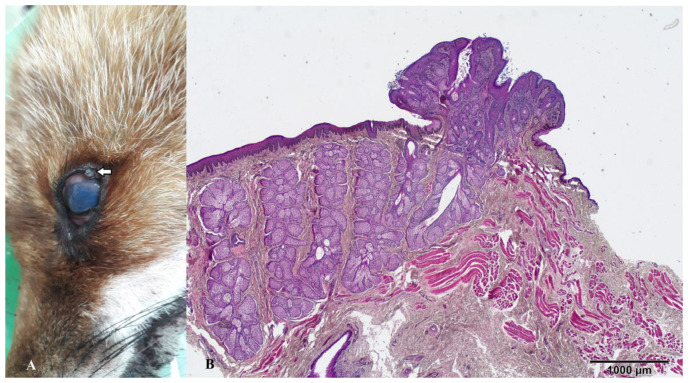
(**A**) Gross appearance of the Meibomian gland adenoma on the upper eyelid (arrow). (**B**) Microscopic appearance of the same lesion depicting a well-circumscribed, benign, intradermal lobular tumor.

**Figure 4 animals-14-00558-f004:**
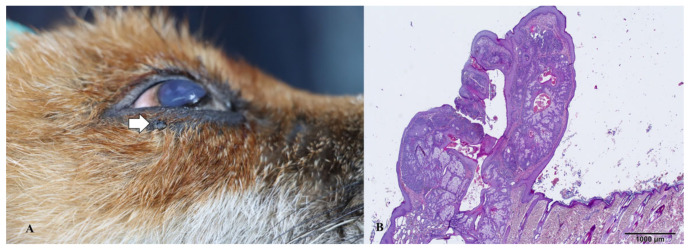
(**A**) Gross appearance of the Meibomian gland adenoma on the lower eyelid (arrow). (**B**) Microscopic appearance of the same lesion depicting a well-circumscribed, benign, intradermal lobular tumor with visible ductal dilation.

**Figure 5 animals-14-00558-f005:**
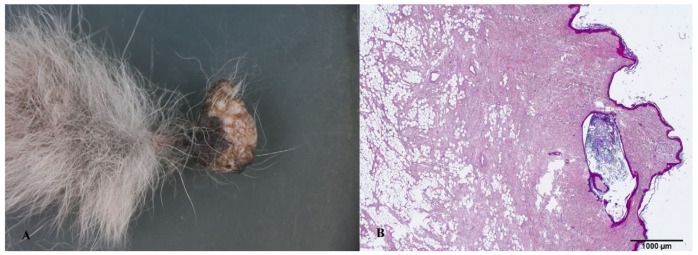
(**A**) Gross appearance of the collagenous hamartoma removed from the abdomen. (**B**) Microscopic appearance of the same lesion with visible bundles of cell-poor collagenous tissue with one dilated hair follicle.

**Figure 6 animals-14-00558-f006:**
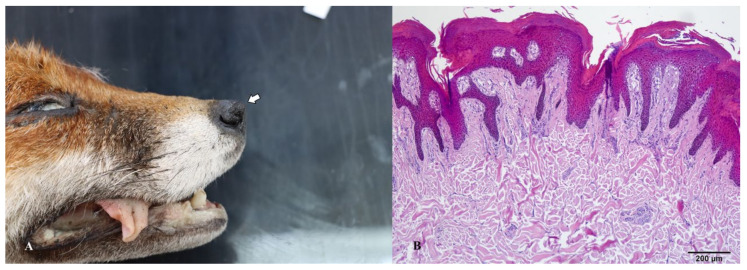
(**A**) Gross appearance of the collagenous hamartoma on the nose apex (arrow). (**B**) Microscopic picture with visible bundles of collagenous tissue and epidermal hyperkeratosis.

**Figure 7 animals-14-00558-f007:**
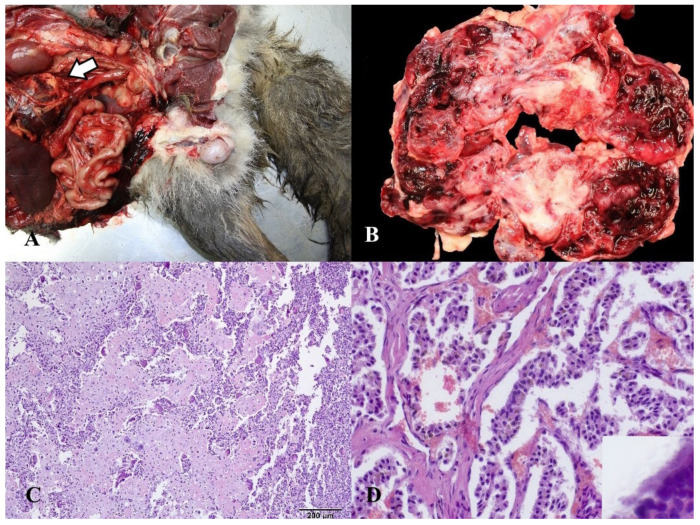
(**A**) Gross position of the teratoma inside the abdomen, between the spleen and kidney (arrow). (**B**) Cross section of teratoma with a central connective tissue and a vascular part with multiple blood spaces on both sides. (**C**) Microscopic appearance of the central part with cartilage formation, osteoid, and multinuclear osteoclasts. (**D**) Microscopic appearance of elongated, prism-shaped epithelial cells with well-developed cilia on their surface (insert photo).

**Table 1 animals-14-00558-t001:** Red foxes with detected tumor and tumor-like lesion; M-male; F- female.

Animals	Cause of Death	Sex	Age	Location of the Tumor/Tumor-like Lesion	Year of Necropsy	Diagnosis
Red fox (*Vulpes vulpes*)	Shot in legal hunt	M	2 years	Intraabdominal	2020	Teratoma
Red fox (*Vulpes vulpes*)	Shot in legal hunt	M	4 years	Upper eyelid	2021	Meibomian gland adenoma
Red fox (*Vulpes vulpes*)	Shot in legal hunt	F	4 years	Ventral abdomen	2022	Collagenous hamartoma
Red fox (*Vulpes vulpes*)	Shot in legal hunt	F	5 years	Lower eyelid	2023	Meibomian gland adenoma
Red fox (*Vulpes vulpes*)	Shot in legal hunt	M	7 years	Nose	2023	Collagenous hamartoma

## Data Availability

The data presented in this study are available on request from the first author.

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
