# Peer review of "Tumor and Tumor-like Lesions in Red Foxes (Vulpes vulpes) from Croatia"

_animals, 2024, doi:10.3390/ani14040558_

Round 1

Reviewer 1 Report

Comments and Suggestions for Authors

The article describes the incidence of tumor and tumor like lesions in red foxes in Croatia. Other similar studies reported in the manuscript already pointed for a low incidence of tumors in this species, so novelty and applicability of the data is limited.

The article requires a minor English revision but is otherwise well described.

The number of cadavers where tumors were found is quite low. The mean age of the animals with tumors is also low but no information was provided related to the overall sample from which the 5 animals with tumors were found. Perhaps the fact that most of the cadavers are from relatively young animals explains why the incidence of tumors is low. An overall overview of ages, sex distribution and causes of death of the remaining animals would complete the manuscript and help on the interpretation of results.

No information on who performed the pathological analysis.

There is no additional information on whether the cause of death was the tumor or if there were other pathological changes in the 5 animals sample.

Comments on the Quality of English Language

Minor english revision is required

Author Response

  1. The article describes the incidence of tumor and tumor like lesions in red foxes in Croatia. Other similar studies reported in the manuscript already pointed for a low incidence of tumors in this species, so novelty and applicability of the data is limited.

Response: We would like to thank the reviewer for their comment, but given that this type of research or data did not previously exist for this part of Europe, we believe it is definitely worth publishing and that it significantly expands our knowledge on the occurrence of tumors in foxes.

  1. The article requires a minor English revision but is otherwise well described.

Response: We have conducted an English language audit.

  1. The number of cadavers where tumors were found is quite low. The mean age of the animals with tumors is also low but no information was provided related to the overall sample from which the 5 animals with tumors were found. Perhaps the fact that most of the cadavers are from relatively young animals explains why the incidence of tumors is low. An overall overview of ages, sex distribution and causes of death of the remaining animals would complete the manuscript and help on the interpretation of results.

Response: We appreciate the reviewer's thorough feedback, which we fully accept. Based on this, we have included a table with age data and graphs showing the causes of death for the fox population in which we found tumors.

  1. No information on who performed the pathological analysis.

Response: According to the reviewer's suggestion in the section Author contributions, we have listed the author who performed the pathological analysis.

  1. There is no additional information on whether the cause of death was the tumor or if there were other pathological changes in the 5 animals sample.

Response: In accordance with the reviewer's comment, we have incorporated the cause of death for foxes with identified tumors into Table 1

Reviewer 2 Report

Comments and Suggestions for Authors

General

The paper describes the occurrence of neoplasms in red foxes submitted for rabies surveillance in Croatia. Only five cases were reported from 1890 animals investigated. As the authors indicate, the relatively low age of the foxes compared to the dog population is mentioned as a possible explanation for the low incidence of neoplasms in foxes compared to pet dogs. Normally, such cases are described as an unusual (coincidental) finding but the fact that these cases were a result of a systemic study using a large sample size makes it a very interesting study worthwhile of being published. Interestingly, the authors never refer in the discussion to the transmissible facial tumors (DFTD) of the Tasmanian Devil (although not a canids species) as an example of a neoplasm with a very high incidence in wildlife (see line 53-54).

Specific:

Line 41: I don’t know if ‘nature’ is a habitat?

Line 45-47: This is a rather strange definition of ‘reservoirs’. The proximity to humans and dogs does not automatically imply that an animal species is a reservoir. If the animal species is a reservoir, living in close proximity to humans or dogs, makes it easier to transmit the disease to the latter two.

Line 46-48: I seriously doubt that these are the most important mortality factors for young and adult foxes. The references used are describing a particular situation (rehabilitation centre and mortality during a mange outbreak) and are not valid for the fox population in general.

Line 49-51: It may be the main cause of death for well supervised pet dogs but also here I doubt that it is the leading cause of death for dogs worldwide, incl. the many free-roaming dogs in the developing countries.

Line 52: I don’t understand why it is ‘prudent’ to study it in foxes.

Line 132: The life span of a free-living red fox is not very long and the average age of 4.4 years of the foxes with tumors is (most likely) much higher than the average age of the fox population, indicating that tumors most often occur in older foxes.

Line 177: A two-year old fox is not considered a juvenile but an adult fox. Juveniles are most of the time defined as 3 – 12 months old.

Line 192-196: This information has already been shared in the introduction, no need to repeat it here.

Author Response

General

  1. The paper describes the occurrence of neoplasms in red foxes submitted for rabies surveillance in Croatia. Only five cases were reported from 1890 animals investigated. As the authors indicate, the relatively low age of the foxes compared to the dog population is mentioned as a possible explanation for the low incidence of neoplasms in foxes compared to pet dogs. Normally, such cases are described as an unusual (coincidental) finding but the fact that these cases were a result of a systemic study using a large sample size makes it a very interesting study worthwhile of being published. Interestingly, the authors never refer in the discussion to the transmissible facial tumors (DFTD) of the Tasmanian Devil (although not a canids species) as an example of a neoplasm with a very high incidence in wildlife (see line 53-54).

Response: We express our gratitude to the reviewer for acknowledging the significance of the provided data for their publication.

Following his instructions, we extended the introduction section to include a brief mention of the transmissible facial tumor in Tasmanian devils, as well as the occurrence of ceruminous gland tumors in island foxes (Urocyon littoralis catalinae) from Santa Catalina Island. As far as we know, these tumors exhibit the highest incidence among wild animals."

Specific:

  1. Line 41: I don’t know if ‘nature’ is a habitat?

Response:  As this word can be confusing, we replaced it with the word „natural habitat“.

  1. Line 45-47: This is a rather strange definition of ‘reservoirs’. The proximity to humans and dogs does not automatically imply that an animal species is a reservoir. If the animal species is a reservoir, living in close proximity to humans or dogs, makes it easier to transmit the disease to the latter two.

Response: We thank the reviewer for his comment, but we do not agree with his opinion because in this sentence, we used the term "reservoir" as defined in the abstract of the cited paper (Ebani, V. Et al., 2022.), which states that foxes from a specific geographic region can be reservoirs for some investigated diseases. Specific diseases whose reservoirs can be foxes we have listed in the discussion in lines 193-195.

  1. Line 46-48: I seriously doubt that these are the most important mortality factors for young and adult foxes. The references used are describing a particular situation (rehabilitation centre and mortality during a mange outbreak) and are not valid for the fox population in general.

Response: Because there are currently no systematic studies in the literature that more accurately describe mortality in foxes, we used these to demonstrate one of the more common causes of death in fox population.

  1. Line 49-51: It may be the main cause of death for well supervised pet dogs but also here I doubt that it is the leading cause of death for dogs worldwide, incl. the many free-roaming dogs in the developing countries.

Response: We appreciate the reviewer's comment, and after careful thought, we added the term "developed countries" to ensure there are no misunderstandings, as well as to be more precise. Baš si mu lepo odgovorio.

  1. Line 52: I don’t understand why it is ‘prudent’ to study it in foxes.

Response: We thank the reviewer for his comment. In the paper, we replaced the word "prudent" with "useful" because a large number of studies on the incidence of tumors in the dog population can help explain and compare the decrease in the incidence of tumors in foxes.

  1. Line 132: The life span of a free-living red fox is not very long and the average age of 4.4 years of the foxes with tumors is (most likely) much higher than the average age of the fox population, indicating that tumors most often occur in older foxes.

Response: We would like to thank the reviewer for his comment, but we still think that this age represents a younger population, because in the submitted fox samples there are also a number of older ones, and in a larger number of cited papers that described tumors in foxes, although most of the foxes were from captivity, the age was much higher, so we can conclude that foxes can live longer than 4.4 years.

  1. Line 177: A two-year old fox is not considered a juvenile but an adult fox. Juveniles are most of the time defined as 3 – 12 months old.

Response: We agree with the reviewer’s comment, and therefore we've corrected that mistake in the paper.

  1. Line 192-196: This information has already been shared in the introduction, no need to repeat it here.

Response: We thank the reviewer for his comment, but we do not agree with that point because in this part of the discussion we expanded the statement from the introduction, in which we only stated that foxes can be reservoirs of disease, while in this part we listed certain viruses, bacteria, and parasites whose reservoirs can be foxes.

Round 2

Reviewer 1 Report

Comments and Suggestions for Authors

All comments were addressed.

Comments on the Quality of English Language

Minor revision required